# NETWORK-AGNOSTIC KNOWLEDGE TRANSFER FOR MEDICAL IMAGE SEGMENTATION

## ABSTRACT

Conventional transfer learning leverages weights of pre-trained networks, but mandates the need for similar neural architectures. Alternatively, knowledge distillation can transfer knowledge between heterogeneous networks but often requires access to the original training data or additional generative networks. Knowledge transfer between networks can be improved by being agnostic to the choice of network architecture and reducing the dependence on original training data. We propose a knowledge transfer approach from a teacher to a student network wherein we train the student on an independent transferal dataset, whose annotations are generated by the teacher. Experiments were conducted on five state-of-the-art networks for semantic segmentation and seven datasets across three imaging modalities. We studied knowledge transfer from a single teacher, combination of knowledge transfer and fine-tuning, and knowledge transfer from multiple teachers. The student model with a single teacher achieved similar performance as the teacher; and the student model with multiple teachers achieved better performance than the teachers. The salient features of our algorithm include: 1) no need for original training data or generative networks, 2) knowledge transfer between different architectures, 3) ease of implementation for downstream tasks by using the downstream task dataset as the transferal dataset, 4) knowledge transfer of an ensemble of models, trained independently, into one student model. Extensive experiments demonstrate that the proposed algorithm is effective for knowledge transfer and easily tunable.

## 1 INTRODUCTION

Deep learning often requires a sufficiently large training dataset, which is expensive to build and not easy to share between users. For example, a big challenge with semantic segmentation of medical images is the limited availability of annotated data (Litjens et al., 2017). Due to ethical concerns and confidentiality constraints, medical datasets are not often released with the trained networks. This highlights the need for knowledge transfer between neural networks, wherein the original training dataset does not need to be accessed. On the other hand, according to the black-box metaphor in deep learning based methods, transferring knowledge is difficult between heterogeneous neural networks. To address these limitations, algorithms were proposed to reuse or share the knowledge of neural networks, such as network weight transfer (Tan et al., 2018), knowledge distillation (Hinton et al., 2015), federated learning (Yang et al., 2019), and self-training (Xie et al., 2020b).

Some conventional algorithms directly transfer the weights of standard large models that were trained on natural image datasets for different tasks (Kang & Gwak, 2019; Motamed et al., 2019; Jodeiri et al., 2019; Raghu et al., 2019). For example, Iglovikov & Shvets (2018) adopted VGG11 pre-trained on ImageNet as the encoder of U-Net for 2D image segmentation. Similarly, the convolutional 3D (Tran et al., 2015), pre-trained on natural video datasets, was used as the encoder of 3D U-Net for the 3D MR (Magnetic Resonance) medical image segmentation (Zeng et al., 2017). Transferring the network weights generally requires adjustments to be made to the architecture of the receiver model, this in turn, limits the flexibility of the receiver network.

Another technique that involves knowledge transfer is federated learning (Yang et al., 2019) ; it has received attention for its capability to train a large-scale model in a decentralized manner without requiring users' data. In general, federated learning approaches adopt the central model to capture the

shared knowledge of all users by aggregating their gradients. Due to the difficulties in transferring knowledge between heterogeneous networks, federated learning often requires all devices, including both the central servers and local users, to use the same neural architecture (Xie et al., 2020a). To our best knowledge, there has been no federated learning system that uses heterogeneous networks.

Knowledge distillation is the process of transferring the knowledge of a large neural network or an ensemble of neural networks (teacher) to a smaller network (student) (Hinton et al., 2015). Given a set of trained teacher models, one feeds training data to them and uses their predictions instead of the true labels to train the student model. For effective transfer of knowledge, however, it is essential that a reasonable fraction of the training examples are observable by the student (Li et al., 2018) or the metadata at each layer is provided (Lopes et al., 2017). Yoo et al. (2019) used a generative network to extract the knowledge of a teacher network, which generated labeled artificial images to train another network. As can be seen, Yoo et al.'s method had to train an additional generative network for each teacher network.

Different from knowledge distillation, self-training aims to transfer knowledge to a more capable model. The self-training framework (Scudder, 1965) has three main steps: train a teacher model on labeled images; use the teacher to generate pseudo labels on unlabeled images; and train a student model on the combination of labeled images and pseudo labeled images. Xie et al. (2020b) proposed self-training with a noisy student for classification, which iterates this process a few times by treating the student as a teacher to relabel the unlabeled data and training a new student. These studies required labeled images for training the student; moreover, they implicitly required that the pseudo-labeled images should be similar in content to the original labeled images.

Inspired by self-training, we propose a network-agnostic knowledge transfer algorithm for medical image segmentation. This algorithm transfers the knowledge of a teacher model to a student model by training the student on a transferal dataset whose annotations are generated by the teacher. The algorithm has the following characteristics: the transferal dataset requires no manual annotation and is independent of the teacher-training dataset; the student does not need to inherit the weights of the teacher, and such, the knowledge transfer can be conducted between heterogeneous neural architectures; it is straightforward to implement the algorithm with fine-tuning to solve downstream task, especially by using the downstream task dataset as the transferal dataset; the algorithm is able to transfer the knowledge of an ensemble of models that are trained independently into one model.

We conducted extensive experiments on semantic segmentation using five state-of-the-art neural networks and seven datasets. The neural networks include DeepLabv3+ (Chen et al., 2018), U-Net (Ronneberger et al., 2015), AttU-Net (Oktay et al., 2018), SDU-Net (Wang et al., 2020), and Panoptic-FPN (Kirillov et al., 2019). Out of the seven datasets, four public datasets involve breast lesion, nerve structure, skin lesion, and natural image objects, and three internal/in-house datasets involve breast lesion (a single dataset with two splits) and thyroid nodule. Experiments showed that the proposed algorithm performed well for knowledge transfer on semantic image segmentation.

## 2 ALGORITHM

The main goal of the proposed algorithm is to transfer the knowledge of one or more neural networks to an independent one, without access to the original training datasets. This section presents the proposed knowledge transfer algorithm for semantic segmentation in Algorithm 1 and its application with fine-tuning for a downstream task in Algorithm 2 (A.4).

The knowledge transfer algorithm first employs one or more teacher models to generate pseudo masks for the transferal dataset and then trains the student model on the pseudo-annotated transferal dataset. For an image $x$ in the transferal dataset $D$, we get the pseudo mask by the weighted average of the teacher models' outputs: $y = \sum_{T_i \in T} w_i \cdot T_i(x)$, where $w_i$ is the weight for the model $T_i$. The output of a model $T_i(x)$ is either a soft mask (pixel value varies from 0 to 1) or a binary mask (pixel value is 0 or 1). Since the ensemble of teacher models is not our primary focus, we simply set the weights equally for all teacher models.

We adopt two constraints to exclude images without salient targets, as shown in Figure 1. The first constraint is on the target pixel number in the pseudo mask, where target pixel indicates the pixel with a value above 0.5. We exclude the image (of size 384 ×384 pixels) if the target pixel number is less than a threshold of 256. The second constraint is on the gray-level entropy of the pseudo

---

**Algorithm 1** Network-Agnostic Knowledge Transfer for Semantic Segmentation

---

**Require:** Teacher set $T$ with trained models, randomly initialized student model $S$
**Require:** Transferal dataset $D$
**Require:** Target pixel number threshold $\alpha$, gray-level entropy threshold $\beta$
 1: **for** each datapoint $x$ in $D$ **do**
 2:    Obtain pseudo mask $y = \sum_{T_i \in T} w_i \cdot T_i(x)$
 3:    $N \leftarrow$ number of target pixels (value above 0.5) in $y$
 4:    $E \leftarrow$ gray-level entropy of $y$
 5:    **if** $N < \alpha$ or $E > \beta$ **then**
 6:        Exclude $x$ from $D$
 7:    **else**
 8:        Update $x$ as $(x, y)$
 9:    **end if**
10: **end for**
11: Train student model $S$ on pseudo-annotated $D$
12: **return** Trained $S$

---

mask. A high entropy value implies the target and the background have no obvious difference and vice versa. We exclude images with a gray-level entropy higher than a threshold of 2.5. We employ the second constraint only in the case that the teacher model outputs a soft mask rather than a binary mask.

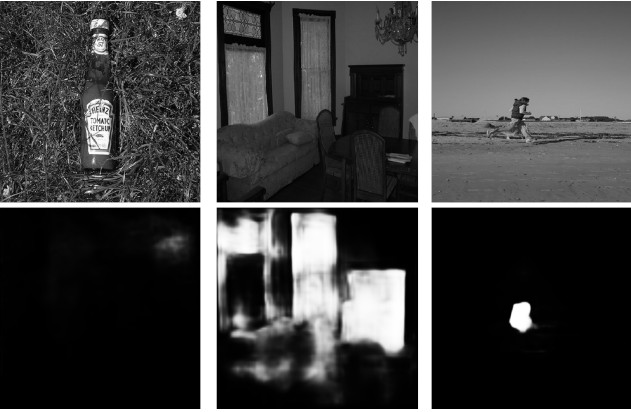

Figure 1: Pseudo mask to target breast lesions in images (converted to grayscale) from PASCAL VOC2012. The top row is original images and the second row is pseudo masks. The first column presents an example with few target pixels. The second column is an example with high gray-level entropy of pseudo mask. The third column shows an example with salient target.

In this algorithm, the teacher models and student model can be independent of each other and do not rely on each other during the training or testing phase. The transferal dataset is independent of the teacher-training datasets, which are not observable for the student model. The student model gains the knowledge of the teacher by the transferal dataset. **As a result, the student model is trained to work well on datasets similar to the teacher-training dataset, while it is not aimed to predict the ground truth of the transferal dataset.**

In the case that a small dataset $D_t$ with ground truth is available for a downstream segmentation task (target task), it is straightforward to further fine-tune the student model that has been trained on pseudo-annotated transferal dataset, as Algorithm 2 shows. The transferal dataset $D$ is independent of the target task, while it can also be a non-annotated dataset from the target task.

## 3    RELATED NEURAL NETWORKS

We posit that if the teacher model is ineffective, a more capable student model can easily achieve similar performance; if the student model, however, is ineffective, the cause cannot be easily identified and the reason can be attributed to the knowledge transfer algorithm and the inherent limitations of the student model. We experimented with five neural networks, all of which have different architectures and have been shown to provide the best outcome on various segmentation tasks.

**DeepLabv3+:** DeepLabv3+ (Chen et al., 2018) is one of Google's latest and best performing semantic segmentation models. It combines the advantages of spatial pyramid pooling with an encoder-decoder structure. Spatial pyramid pooling captures rich contextual information while the decoder path is able to gradually recover object boundaries.

**U-Net:** U-Net (Ronneberger et al., 2015) has been the most widely used network for medical image segmentation. U-Net consists of an encoder-decoder structure with the encoder path extracting rich semantic information, and the decoder path recovering resolution and enabling contextual information.

**AttU-Net:** AttU-Net (Oktay et al., 2018) is a modification of U-Net, where attention gates are used to control the skip connections from the encoder to the decoder. Attention gates make the model focus on target structures by highlighting important features and suppressing irrelevant features.

**SDU-Net:** SDU-Net (Wang et al., 2020) utilizes parallel atrous convolutions with different dilation rates in each layer, which is effective in capturing features in both small and large receptive fields. SDU-Net has demonstrated better performance using merely 40 percent of the parameters in U-Net.

**Panoptic-FPN:** Panoptic-FPN (Kirillov et al., 2019) merges semantic segmentation and object detection, such that each pixel is given a class as in semantic segmentation and each object is given a unique ID as in object detection. Panoptic-FPN provides a more rich and complete segmentation.

## 4    DATASETS

Table 1 presents an overview of the seven image datasets that were used for the experiments. One example for each dataset is presented in Figure 2.

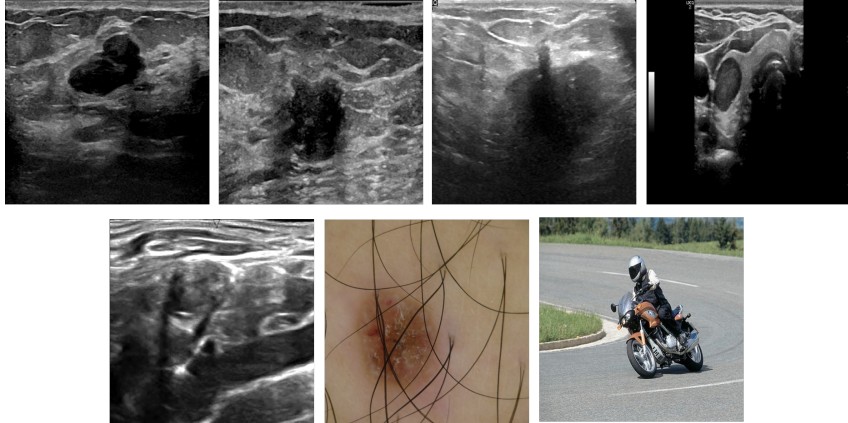

Figure 2: Example for each datasets. From left to right, the top row presents the images from XXX Breast Lesion-1, XXX Breast Lesion-2, Baheya Breast Lesion, and Thyroid Nodule; the second row presents the images from Nerve Structure, Skin Lesion, and PASCAL VOC2012.

**XXX Breast Lesion:** Patients evaluated and referred for Ultrasound (US)-guided biopsy (BI-RADS 4 or 5 mass) at XXX Institute between September 2017 and October 2019 were recruited. Eligible subjects were adult patients with breast masses detected by conventional US imaging. A total of 1385 US images were adopted and manually annotated by two radiologists. These images were randomly assigned to XXX Breast Lesion-1 (n=1237) and XXX Breast Lesion-2 (n=148).

Table 1: Datasets. XXX is used for Anonym.

| Dataset | Image Number | Modality | Data Source |
|---|---|---|---|
| XXX Breast Lesion-1 | 1,237 | Ultrasound | XXX |
| XXX Breast Lesion-2 | 148 | Ultrasound | XXX |
| Baheya Breast Lesion | 697 | Ultrasound | Baheya Hospital |
| Thyroid Nodule | 3,830 | Ultrasound | XXX |
| Nerve Structure | 11,143 | Ultrasound | Kaggle Challenge |
| Skin Lesion | 25,331 | Dermoscopy | ISIC Challenge |
| PASCAL VOC2012 | 17,125 | Natural | VOC Challenge |

**Baheya Breast Lesion:** The Baheya Breast Lesion dataset was released by Al-Dhabyani et al. (2020) and it contains 780 breast US images among women between 25 and 75 years old. Radiologists from Baheya Hospital, Egypt manually annotated the lesion boundaries. The images are categorized into three classes, which are normal (n=133), benign (n=487), and malignant (n=210). In our experiment, we only used images with benign or malignant lesions, totaling 697 images.

**Thyroid Nodule:** A total of 3,830 US images of thyroid nodules were retrospectively collected form the XXX Thyroid Nodule Dataset. Data from patients (more than 18 years old) who underwent US examination before their thyroid biopsy between April 2010 and April 2012 were included. Manual delineation of thyroid nodules by two expert radiologists were the reference standard.

**Nerve Structure:** Hosted on Kaggle, the US Nerve segmentation dataset formed the basis of a competition in segmenting nerve structures in US images of the neck. The dataset includes potential noise, artifacts, and mistakes when labeling the ground truth as well as near-identical images due to the inherent image acquisition technique. In our experiment, we utilized all 11,143 original images.

**Skin Lesion:** The skin-lesion dataset was from a challenge hosted by the International Skin Imaging Collaboration (ISIC) in 2019 where the task was to perform multiclass classification to distinguish dermoscopic images among nine distinct diagnostic categories of skin cancers. We employed the original images of the training set with 25,331 images across 8 classes.

**PASCAL VOC2012:** Pascal VOC2012 challenge dataset is popular for object detection and segmentation tasks. This dataset consists of natural images belonging to 20 different types of objects. We adopted a total of 17,125 original images both for object detection and segmentation.

## 5 EXPERIMENT

Experimental evaluation was conducted in the space of semantic image segmentation. The experiments included: knowledge transfer with a single teacher, combination of knowledge transfer and fine-tuning, and knowledge transfer with multiple teachers. As mentioned in Algorithms 1 and 2, we excluded images without salient targets in their pseudo masks. So we also tested and compared the algorithm's performance with and without image exclusion.

In addition, we evaluated the knowledge transfer capability of each transferal dataset, which is defined as the ratio of the highest Dice score of all student models that are trained on the transferal dataset to the Dice score of the teacher model that generates pseudo masks for the transferal dataset. It is obvious the student and the teacher Dice scores should be calculated on the same test dataset.

### 5.1 KNOWLEDGE TRANSFER WITH A SINGLE TEACHER

We adopted DeepLabv3+ as the teacher model with XXX Breast Lesion-1 as the teacher-training dataset, and employed U-Net, AttU-Net, SDU-Net, and Panoptic-FPN as the student models. We used Baheya Breast Lesion, Thyroid Nodule, Nerve Structure, Skin Lesion, and PASCAL VOC2012 as transferal datasets. The trained teacher processed the transferal datasets to generate pseudo masks, which were used to train the students. We tested the teacher and students on XXX Breast Lesion-2 and Baheya Breast Lesion. The performance was evaluated by computing the Dice score between the model output and manual annotations, i.e., ground truth.

Table 2: Average Dice scores (±std, %) of segmentation on XXX Breast Lesion-2 by student models trained on pseudo-annotated datasets. * indicates image exclusion was employed. KTC indicates knowledge transfer capability.

|  | U-Net | AttU-Net | SDU-Net | Panoptic | KTC |
|---|---|---|---|---|---|
| Baheya Breast Lesion | 72.64±20.93 | 74.72±22.28 | 85.00±14.48 | 81.12±22.52 | 97.66 |
| Baheya Breast Lesion* | 71.74±27.65 | 67.56±32.94 | 82.33±20.89 | 79.95±22.62 | |
| Thyroid Nodule | 66.28±29.23 | 71.17±25.10 | 86.12±14.11 | 82.77±18.70 | 98.94 |
| Thyroid Nodule* | 67.87±30.02 | 75.60±23.48 | 84.39±17.01 | 84.77±16.67 | |
| Nerve Structure | 45.07±37.40 | 58.66±33.61 | 78.53±23.24 | 80.06±23.41 | 93.19 |
| Nerve Structure* | 58.19±32.94 | 57.54±36.30 | 78.92±22.27 | 81.11±21.75 | |
| Skin Lesion | 38.92±22.12 | 34.93±22.16 | 50.10±31.64 | 55.16±33.85 | 81.16 |
| Skin Lesion* | 56.81±28.26 | 53.33±22.18 | 69.17±29.91 | 70.64±28.50 | |
| PASCAL VOC2012 | 61.83±33.61 | 66.18±31.39 | 82.44±18.35 | 79.68±20.73 | 94.72 |
| PASCAL VOC2012* | 70.94±27.40 | 76.22±24.31 | 81.92±19.30 | 81.28±18.97 | |

The teacher model had an average Dice score of 87.04% (±16.01%) on XXX Breast Lesion-2 and an average Dice score of 65.93%(±32.55%) on Baheya Breast Lesion. As shown in Tables 2 and 3, all student model obtained impressive knowledge. For XXX Breast Lesion-2, SDU-Net and Panoptic-FPN obtained Dice scores similar to and even better than the teacher in most of the cases, even though the transferal datasets have different kinds of objects and even different image modalities, such as PASCAL VOC2012. For Baheya Breast Lesion, all students trained on pseudo-annotated Baheya Breast Lesion resulted in higher Dice scores than the teacher model even though the size of Baheya Breast Lesion is much smaller than the teacher-training dataset. The superior performance of the students could be partly due to domain adaptation, which is one of the advantages of our algorithm when the tansferral dataset is from the target task.

The performance of student models was improved by excluding images that had few target pixels or high gray-level entropy from the transferal dataset, especially when the transferal dataset modality was different from the teacher-training dataset. It is because the image of a different modality may result in more pseudo masks with few target pixels or high gray-level entropy, which could be seen from the entropy distribution in Figure 3 (A.2). However, image exclusion did not work that well on Baheya Breast Lesion. It is probably because Baheya Breast Lesion is similar to the teacher-training dataset and of a much smaller size; image exclusion could further harm its representation capability rather than improve its representation quality.

All the transferal datasets have strong knowledge transfer capabilities that are close to and even above 100%, except the Skin Lesion. Although Skin Lesion did not perform as well as the other transferal datasets, its knowledge transfer capability achieved 81.16% and 87.30% according to the two tests. In general it is not a challenge to build a transferal dataset with a strong knowledge transfer capability. At the same time, it is necessary to point out that the four student networks have different learning abilities on the same transferal dataset. However, it is common for different networks to learn differently even on the dataset with manual annotation.

## 5.2 COMBINATION WITH FINE-TUNING

To evaluate the usefulness of knowledge transfer for downstream tasks, we conducted fine-tuning using 50 images with ground truth annotations (from Baheya Breast Lesion and Thyroid Nodule, respectively). Both the teacher model and student models that were trained in the previous experiments were further fine-tuned. For comparison, we also conducted the experiment by training the student models from scratch on these images with ground truth. All the models were tested on the rest of the images of Baheya Breast Lesion and Thyroid Nodule, respectively

The fine-tuned teacher model, DeepLabv3+, resulted in an average Dice score of 69.90%±31.01% on Baheya Breast Lesion and an average Dice score of 66.56%±27.78% on Thyroid Nodule. The results of the student models are presented in Tables 4 and 5. All fine-tuned student models outperformed the models trained from scratch by a large margin. With the same fine-tuning dataset, some of the student models, such as SDU-Net and Panoptic-FPN, performed similar to or even better than

Table 3: Average Dice scores (±std, %) of segmentation on Baheya Breast Lesion by student models trained on pseudo-annotated datasets. * indicates image exclusion was employed. KTC indicates knowledge transfer capability.

| | U-Net | AttU-Net | SDU-Net | Panoptic | KTC |
|---|---|---|---|---|---|
| Baheya Breast Lesion | 65.58±31.22 | 65.14±31.07 | 69.36±30.82 | 67.89±31.93 | 106.66 |
| Baheya Breast Lesion* | 66.73±29.86 | 64.98±31.83 | 69.66±30.31 | 68.84±31.29 | |
| Thyroid Nodule | 47.41±35.20 | 54.63±33.18 | 63.71±32.80 | 62.30±33.27 | 103.90 |
| Thyroid Nodule* | 52.07±35.84 | 57.92±33.87 | 68.50±30.0 | 67.39±30.42 | |
| Nerve Structure | 35.47±36.60 | 42.86±36.92 | 59.62±34.10 | 62.39±32.28 | 94.63 |
| Nerve Structure* | 41.83±34.91 | 44.26±37.92 | 59.41±32.98 | 62.10±32.22 | |
| Skin Lesion | 28.16±22.57 | 27.64±22.77 | 30.17±33.53 | 40.94±37.19 | 87.30 |
| Skin Lesion* | 37.15±29.99 | 37.03±26.36 | 54.54±36.20 | 57.56±35.78 | |
| PASCAL VOC2012 | 39.30±27.48 | 40.62±30.04 | 58.68±30.57 | 57.24±31.82 | 99.38 |
| PASCAL VOC2012* | 52.76±35.19 | 51.52±37.13 | 65.52±31.87 | 63.20±34.35 | |

Table 4: Average Dice scores (±std, %) for student models fine-tuned or trained from scratch on Baheya Breast Lesion. * indicates image exclusion was employed.

| | U-Net | AttU-Net | SDU-Net | Panoptic |
|---|---|---|---|---|
| Trained from Scratch | 37.20±24.65 | 37.15±24.74 | 46.88±28.87 | 52.88±30.31 |
| Baheya Breast Lesion | 59.29±29.04 | 65.31±29.68 | 70.03±31.25 | 70.61±29.64 |
| Baheya Breast Lesion* | 61.45±29.95 | 64.07±30.48 | 70.85±28.88 | 70.97±30.04 |
| Thyroid Nodule | 57.08±28.51 | 60.09±30.04 | 70.48±30.26 | 68.04±29.50 |
| Thyroid Nodule* | 54.06±29.41 | 59.38±29.54 | 71.60±28.95 | 68.60±29.94 |
| Nerve Structure | 56.58±26.94 | 63.21±30.61 | 66.55±31.83 | 71.19±28.29 |
| Nerve Structure* | 57.70±28.95 | 56.76±29.40 | 69.33±29.60 | 68.13±30.21 |
| Skin Lesion | 45.03±26.08 | 43.29±27.52 | 53.45±29.92 | 62.18±31.91 |
| Skin Lesion* | 52.41±27.59 | 51.71±28.41 | 70.62±29.39 | 67.73±29.00 |
| PASCAL VOC2012 | 60.87±30.06 | 62.41±30.57 | 69.93±29.79 | 70.91±28.40 |
| PASCAL VOC2012* | 57.11±29.17 | 61.63±30.24 | 70.83±29.92 | 71.67±29.32 |

the fine-tuned teacher model. This further verified that the effect of knowledge transfer between heterogeneous neural networks and demonstrate that the proposed algorithm can be well applied to downstream task.

## 5.3 KNOWLEDGE TRANSFER WITH MULTIPLE TEACHERS

We also tested if it was possible to transfer the knowledge of multiple weak independent teachers to build a stronger student. We adopted U-Net, AttU-Net, SDU-Net, and Panoptic-FPN as the teacher models and employed DeepLabv3+ as a student model. We train the teacher models on XXX Breast Lesion-1 for a few epochs to ensure under-fitting, thereby resulting in weak teachers. Baheya Breast Lesion was used as the transferal dataset. The trained teachers processed each image in the transferal dataset, with the average soft mask as pseudo mask. Then we trained the student model on the pseudo-annotated transferal dataset.

We compared the performance of the teacher models and the student model on XXX Breast Lesion-2 and Baheya Breast Lesion (Table 6). The student model outperformed all the teacher models on both test datasets. This is primarily due to fact that the student model actually learned from the ensemble of the teacher models, rather than any individual weak teacher model. Since the process of generating ensemble networks is not the primary focus of this study, we simply average their outputs. With an advanced ensemble, we may be able to train the student model better.

Table 5: Average Dice scores (±std, %) for student models fine-tuned or trained from scratch on Thyroid Nodule. * indicates image exclusion was employed.

|  | U-Net | AttU-Net | SDU-Net | Panoptic |
|---|---|---|---|---|
| Trained from Scratch | 34.02±23.89 | 35.85±24.51 | 46.00±27.08 | 48.96±26.92 |
| Baheya Breast Lesion | 50.93±27.95 | 41.28±47.74 | 57.99±30.00 | 64.92±28.20 |
| Baheya Breast Lesion* | 47.65±28.37 | 50.51±28.40 | 56.80±29.94 | 63.59±29.35 |
| Thyroid Nodule | 50.16±29.60 | 47.91±30.37 | 59.26±32.55 | 64.73±29.9 |
| Thyroid Nodule* | 55.41±27.69 | 57.18±28.82 | 60.15±31.34 | 65.65±29.32 |
| Nerve Structure | 44.93±27.00 | 45.51±30.80 | 53.05±33.07 | 65.37±28.23 |
| Nerve Structure* | 47.1±26.66 | 45.650±28.98 | 53.92±28.57 | 63.26±29.55 |
| Skin Lesion | 41.32±23.37 | 34.41±24.68 | 49.97±29.69 | 54.30±30.29 |
| Skin Lesion* | 43.21±25.98 | 43.44±28.61 | 55.91±30.04 | 56.00±28.25 |
| PASCAL VOC2012 | 41.93±26.60 | 48.59±27.08 | 57.45±28.35 | 63.09±27.31 |
| PASCAL VOC2012* | 44.81±29.13 | 51.47±29.43 | 59.85±29.90 | 64.63±28.35 |

Table 6: Average Dice scores (±std, %) for knowledge transfer with multiple teacher models.

|  | U-Net | AttU-Net | SDU-Net | Panoptic | DeepLabv3+ |
|---|---|---|---|---|---|
| XXX Br. Lesion-2 | 67.47±27.98 | 76.99±21.96 | 73.63±18.65 | 71.49±27.44 | 78.71±18.6 |
| Baheya Br. Lesion | 49.30±34.33 | 49.22±34.25 | 48.75±32.43 | 48.95±34.01 | 58.40±31.05 |

## 6 DISCUSSION

The salient features of our algorithm include: no need for original training data or extra generative networks; facilitating knowledge transfer between networks with different architectures; ease of implementation with fine-tuning to solve downstream task especially when using the downstream task dataset as the transferal dataset; capability of knowledge transfer from multiple models, trained independently, into one student model. The algorithm can also perform domain adaptation without extra computation in the case that the downstream task dataset is used as the transferal dataset.

To the best of our knowledge, this is the first demonstration of image segmentation using pseudo annotations only. A transferal dataset with a large number of images of various contents has higher possibility to capture rich targets in the pseudo masks, but it may also include many images without salient targets. Future work may include optimization of a single transferal dataset or a combination of multiple transferal datasets to build a better one. Different models may have different abilities to learn from the pseudo-annotated transferal dataset. Understanding the differences in student model learning on pseudo-annotated transferal dataset and manually annotated dataset can help generate confidence in the proposed algorithm. Further study on knowledge transfer between neural networks with different input data structures or different number of prediction classes is also warranted. Another possible direction includes extending the ensemble of teacher models to produce pseudo masks.

Medical imaging applications require interpretable algorithms which generate confidence in the clinical user. Knowledge transfer employs pseudo annotation for training; these pseudo annotations have no physical meaning. It is imperative to examine and quantify the interpretability of student model before deploying models clinically.

## 7 CONCLUSION

Knowledge transfer can be achieved between neural networks having different architectures and using an independent transferal dataset. Knowledge transfer from multiple networks improves the performance of the student network over the teacher networks.

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

## A  APPENDIX

### A.1  DATA PRE-PROCESSING AND AUGMENTATION

In addition to pre-processing, we employed five methods for data augmentation.

**Pre-processing:** All the images were resized to $384 \times 384$ and the color images in Skin Lesion and PASCAL VOC2012 were converted into gray-scale images.

**Random Cropping:** A percentage of consecutive rows and columns were cropped from the original image. The percentage was a random value in the range [70%, 100%] and [80%, 100%] for segmentation and classification, respectively.

**Horizontal Flipping:** The image was reversed horizontally, that is, from left to right.

**Random Rotation:** The image was rotated around its center. The rotation angle was a random value in the range [-45°, 45°] and [-30°, 30°] for segmentation and classification, respectively.

**Gamma Adjustment**: Gamma correction was conducted on an image by the gamma value, which was randomly sampled in the range [0.7, 1.3].

**Contrast Adjustment:** The image contrast of an image was adjusted by the contrast factor, which was a random value in the range [0.7, 1.3].

### A.2  DATA DISTRIBUTION AND SELECTION

We used DeepLabv3+ that was trained on XXX Breast Lesion-1 to process the images in the five transferal datasets including Baheya Breast Lesion, Thyroid Nodule, Nerve Structure, Skin Lesion, and PASCAL VOC2012, and obtained the pseudo mask. Figure 3 presents the gray-level entropy distribution of the pseudo mask for each transferal dataset.

By excluding images without obvious targets, where the target pixel number is less than 256 and the gray-level entropy is over 2.5, we could reduce the dataset to a smaller one. Table 7 presents the comparison of image numbers with and without image exclusion.

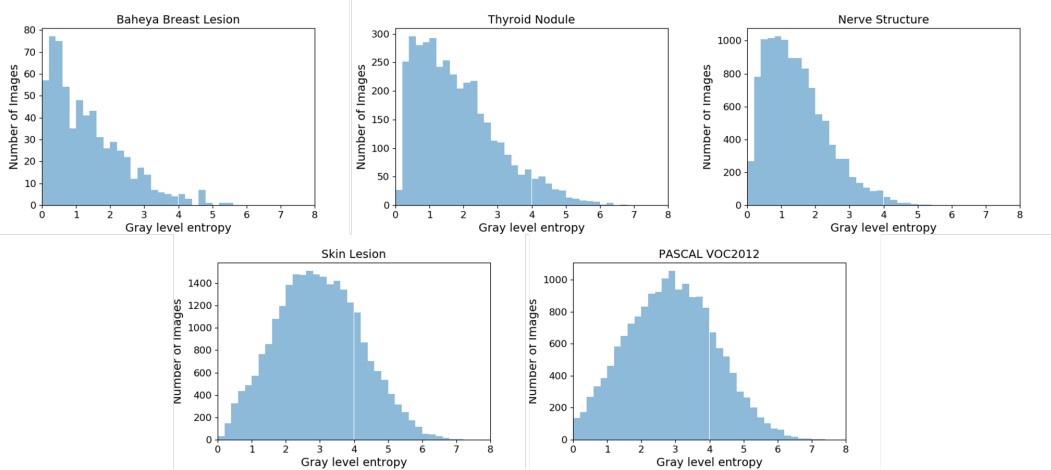

Figure 3: Image distribution with respect to the entropy. From left to right, the top row presents the result for Baheya Breast Lesion, Thyroid Nodule, and Nerve Structure; the second row presents the examples for Skin Lesion and PASCAL VOC2012.

Table 7: Image number of each transferal dataset before and after the exclusion of images without salient target in pseudo masks.

| Transferal Dataset | Images before Exclusion | Images after Exclusion |
|---|---|---|
| Baheya Breast Lesion | 697 | 525 |
| Thyroid Nodule | 3,830 | 2,790 |
| Nerve Structure | 11,143 | 8,977 |
| Skin Lesion | 25,331 | 6,960 |
| PASCAL VOC2012 | 17,125 | 5,503 |

### A.3 TRAINING DETAILS

Two common loss functions were adopted with the same weight for the segmentation tasks, the Dice loss function and binary cross entropy loss (BCE). The Dice loss is defined as:

$$L_{dice} = 1 - \frac{2\sum p_{h,w} \cdot \hat{p}_{h,w} + \epsilon}{\sum p_{h,w} + \sum \hat{p}_{h,w} + \epsilon} \tag{1}$$

where $(h, w)$ represents the pixel coordinate, $p_{h,w} \in \{0, 1\}$ is the mask ground truth, $p_{h,w} = 1$ indicates the pixel belonging to the target, $0 \leq \hat{p}_{h,w} \leq 1$ is the prediction probability for the pixel belonging to the target, $\epsilon$ is a small real number.

The binary cross entropy loss is defined as:

$$L_{bce} = -\frac{1}{N} \sum p_{h,w} \cdot log(\hat{p}_{h,w}) + (1 - p_{h,w}) \cdot log(1 - \hat{p}_{h,w}) \tag{2}$$

where $(h, w)$ represents the pixel coordinate, $p_{h,w} \in \{0, 1\}$ is the mask ground truth, $p_{h,w} = 1$ indicates the pixel belonging to the target, $0 \leq \hat{p}_{h,w} \leq 1$ is the prediction probability for the pixel belonging to the target, N is the number of pixels.

Adam optimizer was used for training all the networks. The learning rate was set to 0.0001, and all the other parameters were set to the default values in PyTorch. The batch size was set to 4. The epoch number was set to 500 for the training on the teacher-training dataset. Inversely proportional to the image number, the epoch numbers were set to 500, 175, 60, and 30 to for the training on pseudo-annotated Baheya Breast Lesion, Thyroid Nodule, Nerve Structure, and Skin Lesion. Moreover, the training epoch number for fine-tuning was set to 50.

### A.4  ALGORITHM OF KNOWLEDGE TRANSFER COMBINED WITH FINE-TUNING

Algorithm 2 combines knowledge transfer and fine-tuning for a downstream task of semantic image segmentation. In the algorithm, the transferal dataset is independent of the downstream task, but if it is from the downstream task, the algorithm can benefit from domain adaptation without extra computation.

---

**Algorithm 2** Knowledge Transfer Combined with Fine-tuning for Semantic Segmentation

---

**Require:** Teacher set $T$ with trained models, randomly initialized student model $S$
**Require:** Transferal dataset $D$, target task dataset $D_t$ with annotation
**Require:** Target pixel number threshold $\alpha$, gray-level entropy threshold $\beta$
 1: **for** each datapoint $x$ in $D$ **do**
 2:     Obtain pseudo mask $y = \sum_{T_i \in T} w_i \cdot T_i(x)$
 3:     $N \leftarrow$ number of target pixels (value above 0.5) in $y$
 4:     $E \leftarrow$ gray-level entropy of $y$
 5:     **if** $N < \alpha$ or $E > \beta$ **then**
 6:         Exclude $x$ from $D$
 7:     **else**
 8:         Update $x$ as $(x, y)$
 9:     **end if**
10: **end for**
11: Train student model $S$ on pseudo-annotated $D$
12: Fine-tune student model $S$ on target task dataset $D_t$
13: **return** Trained student model $S$

---

