# OpenReview forum: "Network-Agnostic Knowledge Transfer for Medical Image Segmentation"
_ICLR.cc/2021/Conference — Reject_

### Official Review · AnonReviewer1 · 2020-10-15
**The proposed idea is not novel**

**Rating:** 3
**Confidence:** 4

**Review:**

The paper describes a  knowledge transfer technique based on  training a student network using annotation creating by  a teacher network. This is actually not a summary of the method but  the method itself. Most the the rest of the paper is devote do describe experiment details.

The idea is  well known in machine learning community see e.g. Distilling the Knowledge in a Neural Network by Hinton. where is used to transfer knowledge from a huge network to a small network. Hence, there is not much novelty in the paper.

Although the method is very simple it is difficult the follow the experimental results. It is written in a very unclear way.
Do you use step 3 in the experiments?

what is your conclusion regarding parameter fine tuning vs. your approach?

Over all the paper is more suitable for a medical imaging conference than  fro a general deep learning conference.

---

> ### Author Response · Authors · 2020-11-23
> **Response to Reviewer 1**
>
> ~~~
> Reviewer's comment: The paper describes a knowledge transfer technique based on training a student network using annotation creating by a teacher network. This is actually not a summary of the method but the method itself. Most the the rest of the paper is devote do describe experiment details.
> ~~~
> We sincerely thank the reviewer for taking the time to review our paper.
> We have revised the paper extensively.  Meanwhile, we also highlighted our major contribution as well as the novelty of this article.
>
> By the way, the reviewer describes the main idea of the algorithm correctly. Please find our detailed explanation below.
>
> ~~~
> Reviewer's comment: The idea is well known in machine learning community see e.g. Distilling the Knowledge in a Neural Network by Hinton. where is used to transfer knowledge from a huge network to a small network. Hence, there is not much novelty in the paper.
> ~~~
>
> We  surveyed the work on knowledge  distillation  (Hinton  et  al.,  2015)  and some related studies (Yoo et al., 2019; Lopes et al., 2017). The novelty of our algorithm can be seen at least from four aspects.
>
> -  **The main novelty is that our algorithm does not require any teacher-training data, metadata, or additional generative network**.  Previous studies trained student models heavily relying on the training dataset of the teacher model (Hinton et al.,  2015) , metadata (Lopes et al., 2017), or additional generative networks (Yoo et al., 2019).  As pointed out by Yoo et al. (2019) and Lopes et al. (2017), Hinton’s knowledge distillation (Hinton et al., 2015) needed to access the teacher training dataset (labeled or unlabeled). Lopes et al. (2017) required producing metadata during training and the student model had to be trained on metadata instead.  Yoo et al. (2019) employed an additional generative network to generate artificial dataset for training the student model, so that the performance of the generative network also affected the training of the student network. The generative network was coupled with the teacher network, so that it is necessary to design and train a generative network for each teacher model, which will increase the computation and challenge. It would be even more challenging if there are multiple teacher models. Since the teacher-training dataset, metadata, and generative network was coupled with the teacher network, they limited the application of these algorithms or increased the computation burden to train the student network.
>
> - Our algorithm transfers knowledge between heterogeneous networks of semantic segmentation, while these well-known knowledge distillation studies focused on classification.
>
> - Our transferal dataset (used to train the student model) is allowed to be much different from the teacher training dataset, even of different image modalities.  For example, we used PASCAL VOC2020 as transferal dataset to transfer the ultrasound segmentation knowledge, and it worked well.
>
> - Last but not least, our algorithm is really simple and easy to implement.
>
> We also highlighted the salient features of our algorithm in  **Abstract**-- “The salient features of our algorithm include: 1) no need for original training data or generative networks, 2) knowledge transfer between different architectures, 3) ease of implementation for downstream tasks by using the downstream task dataset as the transferal dataset, 4) knowledge transfer of an ensemble of models, trained independently, into one student model.”
>
> ~~~
> Reviewer's comment: Although the method is very simple it is difficult the follow the experimental results. It is written in a very unclear way. Do you use step 3 in the experiments?
> ~~~
> To make it easy to follow, we carefully revised the paper, especially the experiment section. For the convenience of the reviewer, we would like to make a brief summary of our study:
> - Our main algorithm is on knowledge transfer from a teacher model to an student model that is independent in architecture (as the reviewer described in the first comment), which is Algorithm 1.
> - If a small dataset with ground truth is provided, the knowledge transfer algorithm can be used together with fine-tuning (Algorithm 2).
> - In the revision, we described the two algorithms with more details, separately .
>
> We are looking forward to more comments and suggestions from the reviewer. We will try our best to improve the paper.

---

### Official Review · AnonReviewer3 · 2020-10-28
**Interesting direction but lacks positioning in related work and clearer experimentation**

**Rating:** 4
**Confidence:** 3

**Review:**

The paper proposes to use student-teacher training as a way of knowledge transfer between neural networks with different architectures without access to the source data. Instead the authors propose to use a separate dataset to transfer the knowledge of the teacher network and a potential different dataset for fine-tuning. The paper evaluates their method with various segmentation architectures by pretraining a DeepLab v3+ on an internal breast lesion dataset and testing transfer and fine-tuning using different medical datasets. The authors find that knowledge transfer performs similar to regular transfer learning in most combinations of datasets.

I believe that the paper tackles an interesting scenario of transferring knowledge from a fixed pre-trained network to a potentially different application without access to the original training data and sets up an extensive set of combinations of target tasks, student network architectures and transferral dataset (called dataset agent in the paper).

However, I cannot recommend the paper for acceptance in its current form. Most importantly the paper seems to be lacking proper positioning within the space of knowledge distillation and student-teacher training which leads to an unclear message about the novelty of the paper. Student-teacher training for knowledge distillation is not novel and the message that it is possible to transfer knowledge using student-teacher training  is unsurprising given that it has been shown before that you can transfer knowledge without any observed data (e.g. KegNet: Knowledge Extraction with No Observable Data. Yoo et al. NeurIPS 2019).
Further, the experiments do not contribute any new insights about how to chose the best student network nor which transferral dataset to use even though the introduction refers to unsuitability of certain pretraining tasks for a different target task. The first example (Table 2) seems to be using an internal dataset that randomly has been split into train/val/test splits and therefore resembles no real transfer learning task and it seems unsuprising that the 'direct learning' approach yields the best results. The second example (Table 3) using the Baheya breast lesion dataset seems to tackle the problem of unsupervised domain adaptation rather than transfer learning: the teacher and target dataset both tackle breast lesion segmentation on ultrasound images. Here it could be that using the target dataset as transferral dataset might help to adjust batch statistics for potential normalisation layers to improve the performance. This leaves the last two examples as only real transfer learning experiments. Lastly, the whole setup assumes that the input and output space of the student and teacher network are always the same, while it is argued that this approach allows for flexibility in difference of network architectures between the student and teacher network. However, semantic segmentation tasks in medical imaging often appear with various numbers of classes and 'input channels' requiring more advanced knowledge distillation -- this could be an interesting problem to tackle in a later version of this work.

Further comments:
- Data preprocessing: What preprocessing are you using for the training of the networks? How are you handling different shapes of the images? Are the segmentation algorithms trained on the full images or on patches?
- Transferral datasets: Looking at the images in Fig. 2, it seems that even the same modality ultrasound images show different sorts of image artefacts - do you clean those at all? Do you think domain shift might be something that's interacting with your setup?
- Do you have any theoretical or intuitive justification why you would want to perform knowledge transfer using unrelated data (skin lesion / different anatomy etc)? Why should this be better than using no data at all or regular computer vision datasets? Do you think the number of training examples for transfer matters?
- What do example segmentations look like? Are there similar shapes for different datasets? Does the network also learn some sort of shape prior? (see Oktay, et al. Anatomically constrained neural networks (ACNNs): application to cardiac image enhancement and segmentation. 2017)
- Which loss did you use: CE / Dice-loss or a combination of the two?
- The term dataset agent is already used in the abstract and is not very clear - I'd personally find something like 'transfer[ral] dataset' easier to grasp.
- Introduction, first paragraph: 'black-[space]box' -> 'black-box'
- Introduction: 'network(teacher)' -> 'network (teacher)'
- What's a latent dataset? I would rather simply refer to 'learned representations' or 'knowledge'
- 'XXX' is already used in the introduction and not explained - I would simply refer to 'internal / in-house datasets'. Also, note the comment on the breast lesion dataset only being a single dataset with different splits.
- I have not seen the term 'educated' in reference to neural networks before - it would be more common to say 'trained'.
- Section 5.3.2): You mention that the networks trained from scratch have poor performance because of the small tuning dataset - I guess you are referring to the small training set?
- Another potential reference for knowledge transfer for medical imaging could be Kuzina, et al. Bayesian Generative Models for Knowledge Transfer in MRI Semantic Segmentation Problems. 2019
- As this work is relatively application-specific it might be better suited for one of the more medically inclined venues like MIDL, MICCAI, ...

---

> ### Author Response · Authors · 2020-11-23
> **Response to Reviewer 3 -- 1/4**
>
> We thank the reviewer their insightful comments. Following these suggestions, we have carefully and extensively improved the paper.
>
> ~~~
> Reviewer's comment: Most importantly the paper seems to be lacking proper positioning within the space of knowledge distillation and student-teacher training which leads to an unclear message about the novelty of the paper. Student-teacher training for knowledge distillation is not novel and the message that it is possible to transfer knowledge using student-teacher training is unsurprising given that it has been shown before that you can transfer knowledge without any observed data (e.g. KegNet: Knowledge Extraction with No Observable Data. Yoo et al. NeurIPS 2019).
> ~~~
> As part of our revision, we reviewed all reviewers' recommended papers as well as some other related papers. Accordingly, we updated the introduction section to better position our algorithm within the space of knowledge distillation and student-teacher training.
>
> Here we introduce representative algorithms on knowledge distillation (Hinton  et  al.,  2015; Yoo et al., 2019; Lopes et al., 2017).  These algorithms needed to train student models relying on the training dataset of the teacher model (Hinton et al., 2015), metadata (Lopes et al., 2017), or additional generative networks (Yoo et al., 2019).  As pointed out by Yoo et al. (2019) and Lopes et al. (2017), Hinton’s knowledge distillation (Hinton et al., 2015) needs to access the teacher-training dataset (labeled or unlabeled). Lopes et al. (2017) required producing metadata during training and the student model had to be trained based on this metadata.  Yoo et al. (2019) employed an additional generative network to generate artificial dataset for training the student model. Therefore, it was necessary to design and train a generative network for each teacher model, which increased the computation burden and finally determined the performance of the student network. This process would be even more challenging if there were multiple teacher models. **Since the teacher training dataset, metadata, and generative network were coupled with the teacher network, they limited the application of these algorithms. Comparatively,  our algorithm is simple but effective, does not rely on additional generative networks and does not have any requirements on the teacher-training data or metadata.**
>
> We showed that the transferal dataset was allowed to be different from the teacher training-dataset. For example, we used PASCAL VOC2020 as transferal dataset to transfer the ultrasound segmentation knowledge, and it worked well. Even though our algorithm is straightforward, it was able to solve a particularly challenging problem (even to complex algorithms such as (Yoo et al., 2019)).
>
> Our revised paper presents the explanation in **Section 1**-- “Knowledge distillation is the process of transferring the knowledge of a large neural network or an ensemble of neural networks (teacher) to a smaller network (student) (Hinton et al., 2015). Given a set of trained teacher models, one feeds training data to them and uses their predictions instead of the true labels to train the student model. For effective transfer of knowledge, however, it is essential that a reasonable fraction of the training examples is observable by the student (Li et al., 2018) or the metadata at each layer is provided (Lopes et al., 2017). Yoo et al. (2019) used a generative network to extract the knowledge of a teacher network, which generated labeled artificial images to train another network. As can be seen, Yoo et al.’s method had to train an additional generative network for each teacher network.”
>
> **Reference**
>
> *Geoffrey Hinton, Oriol Vinyals, and Jeff Dean. Distilling the knowledge in a neural network. arXiv preprint arXiv:1503.02531, 2015.*
>
> *Raphael Gontijo Lopes, Stefano Fenu, and Thad Starner. Data-free knowledge distillation for deep neural networks. arXiv preprint arXiv:1710.07535, 2017.*
>
> *Jaemin Yoo, Minyong Cho, Taebum Kim, and U Kang. Knowledge extraction with no observable data. In Advances in Neural Information Processing Systems, pp. 2705–2714, 2019.*

---

> ### Author Response · Authors · 2020-11-23
> **Response to Reviewer 3 -- 2/4**
>
> ~~~
> Reviewer's comment: Further, the experiments do not contribute any new insights about how to choose the best student network nor which transferral dataset to use even though the introduction refers to unsuitability of certain pretraining tasks for a different target task.
> ~~~
>
> **Response about transferal datasets:**
> - We  revised the algorithm section with details about the image selection process (based on the pseudo mask).  Please find our method for image selection  in   **Section 2**--“We adopt two constraints to exclude images without salient targets, as shown in Figure 1. The first constraint is on the target pixel number in the pseudo mask, where target pixel indicates the pixel with a value above 0.5. We exclude the image (the resolution size is 384×384) if the target pixel number is less than a threshold of 256. The second constraint is on the gray-level entropy of the pseudo mask. A higher value implies the target and the background have no obvious difference and vice versa. We exclude images with a gray-level entropy higher than a threshold of 2.5. We only employ the second constraint in the case that the teacher model outputs a soft mask rather than a binary mask.”
>
> - We conducted further experiments which demonstrated that most datasets, even those from different modalities, could work well for knowledge transfer.  We discussed the selection of transferal datasets in **Section 6**--"A transferal dataset with a large number of images with varied content has a higher possibility of capturing rich targets in the pseudo masks, but it may also include many images without salient targets. Future work includes optimizing a single transferal dataset or a combination of multiple transferal datasets to build a better aggregate."
>
> **Response about student models:**
> -  It is true that different models may have different abilities to learn from the pseudo-annotated transferal dataset, while it is similar to models learning on dataset with ground truth (e.g. manual annotations). Understanding the differences in student model learning on pseudo-annotated transferal dataset and manually annotated dataset can help generate confidence in the proposed algorithm. As a novel algorithm, there must be many avenues that can improve and extend the study. We leave the topic on how to build a student network to learn well on pseudo-annotated transferal dataset to our future study.
>
> - We condensed the discussion in **Section 6**--“Different models may have different abilities to learn from the pseudo-annotated transferal dataset. Understanding the differences in student model learning on pseudo-annotated transferal dataset and manually annotated dataset can help generate confidence in the proposed algorithm.”

---

> ### Author Response · Authors · 2020-11-23
> **Response to Reviewer 3 -- 3/4**
>
> ~~~
> Reviewer's comment: The first example (Table 2) seems to be using an internal dataset that randomly has been split into train/val/test splits and therefore resembles no real transfer learning task and it seems unsuprising that the 'direct learning' approach yields the best results. The second example (Table 3) using the Baheya breast lesion dataset seems to tackle the problem of unsupervised domain adaptation rather than transfer learning: the teacher and target dataset both tackle breast lesion segmentation on ultrasound images. Here it could be that using the target dataset as transferral dataset might help to adjust batch statistics for potential normalisation layers to improve the performance. This leaves the last two examples as only real transfer learning experiments.
> ~~~
>
> Our main algorithm is on knowledge transfer (Algorithm 1). However, if a small dataset with ground truth of the downstream task is given, our knowledge transfer can be used together with fine-tuning (Algorithm 2).
>
> In the revision, we removed experiments (direct learning) that might confuse the reviewer and lacked relevance. Table 2 and Table 3 aim to show that student models can transfer the knowledge of the teacher model by various transferal datasets without manual annotation. For example, Panoptic FPN was trained on pseudo-annotated PASCAL VOC2012* from scratch, and it could segment the XXX Breast Lesion-2 with an average Dice score of 81.28±18.97 (Table 2). Panoptic-FPN achieved a 94.72% Dice score of the teacher model.  **Note that all student models were only trained on pseudo-annotated transferal datasets, and they had no access to the teacher-training dataset; the transferal dataset, such as PASCAL VOC2012, can be totally different from the test datasets (XXX Breast Lesion-2 and Baheya Breast Lesion).**
>
>  It is true if the test dataset and the transferal dataset are of the same distribution, our algorithm can achieve domain adaptation without extra computation. This is actually one of the advantages of our algorithm, especially when the downstream task is known and the downstream task dataset is used as the transferal dataset. However, it doesn’t mean our algorithm is only data adaptation. As we can see from Table 3, if we used a dataset of different modality, PASCAL VOC2012*, as the transferal dataset, the student models continued to perform well.  Table 4 and Table 5 show the results by combining knowledge transfer and fine-tuning. It is can be viewed as a kind of transfer learning, with the important point that the knowledge transfer occurs between heterogeneous networks without explicit access to large training datasets with ground truth labels.
>
>
>
>
>
> ~~~
> Reviewer's comment: Lastly, the whole setup assumes that the input and output space of the student and teacher network are always the same, while it is argued that this approach allows for flexibility in difference of network architectures between the student and teacher network. However, semantic segmentation tasks in medical imaging often appear with various numbers of classes and 'input channels' requiring more advanced knowledge distillation -- this could be an interesting problem to tackle in a later version of this work.
> ~~~
>
> We agree that it is one limitation of this work, that we only conducted experiments on networks that have the same input and output space. It would be interesting to further study the knowledge transfer between networks with different input data structure or different number of prediction classes. We have pointed it out in the discussion of the revision.  Please refer to **Section 6** --“Further study on knowledge transfer between neural networks with different input data structures or different number of prediction classes is also warranted.”

---

> ### Author Response · Authors · 2020-11-23
> **Response to Reviewer 3 -- 4/4**
>
> ~~~
> Reviewer's comment: Data preprocessing: What preprocessing are you using for the training of the networks? How are you handling different shapes of the images? Are the segmentation algorithms trained on the full images or on patches?
> ~~~
>
> We introduced data preprocessing in the appendix of the revision (**Section A.1**):
>
> In addition to pre-processing, we employed five methods for data augmentation.
>
> - **Pre-processing:** All the images were resized to 384×384 and the color images in Skin Lesion and PASCAL VOC2012 were converted into gray-scale images.
> - **Random Cropping:** A percentage of consecutive rows and columns were cropped from the original image. The percentage was a random value in the range [70%, 100%] and [80%, 100%] for segmentation and classification, respectively.
> - **Horizontal Flipping:** The image was reversed horizontally, that is, from left to right.
> - **Random Rotation:** The image was rotated around its center. The rotation angle was a random value in the range [-45◦, 45◦] and [-30◦, 30◦] for segmentation and classification, respectively.
> - **Gamma Adjustment:** Gamma correction was conducted on an image by the gamma value, which was randomly sampled in the range [0.7, 1.3].
> - **Contrast Adjustment:** The image contrast of an image was adjusted by the contrast factor, which was a random value in the range [0.7, 1.3].”
>
> ~~~
> Reviewer's comment: Transferral datasets: Looking at the images in Fig. 2, it seems that even the same modality ultrasound images show different sorts of image artefacts - do you clean those at all? Do you think domain shift might be something that's interacting with your setup?
> ~~~
>
> We did not remove or revise any images due to artifacts and we did not take domain shift into consideration.
> We are not sure if we understand this question correctly. Please let us know if you think our answer is confused.
>
> ~~~
> Reviewer's comment: Do you have any theoretical or intuitive justification why you would want to perform knowledge transfer using unrelated data (skin lesion / different anatomy etc)? Why should this be better than using no data at all or regular computer vision datasets? Do you think the number of training examples for transfer matters?
> ~~~
>
> By using unrelated data, we want to show that the transferal dataset is quite easy to build.
> We are not sure of what the reviewer mean by “using no data”. We guess the reviewer refers to Yoo et al. (2019). Although Yoo et al. (2019) claimed they used no data, they needed an **additional** generative network to produce artificial training data. Actually, the generative network itself can be a challenge, as each teacher network needs to train the generative network specially and different tasks may need different generative neural architectures.
>
> ~~~
> Reviewer's comment: What do example segmentations look like? Are there similar shapes for different datasets? Does the network also learn some sort of shape prior? (see Oktay, et al. Anatomically constrained neural networks (ACNNs): application to cardiac image enhancement and segmentation. 2017)
> ~~~
>
> No. They do not have to be similar and they can be totally visually different.
> In the revision, we adopted a natural image dataset, PASCAL VOC2012, as the transferal dataset and it also worked well.
>
> ~~~
> Reviewer's comment: Which loss did you use: CE / Dice-loss or a combination of the two?
> ~~~
> We used both CE and Dice loss  with the same weight.  Please refer to our revision in **Section A.3**--“Two common loss functions were adopted with the same weight for the segmentation tasks”
>
> Finally, we greatly appreciate all the other detailed comments the reviewer provided. These comments are very helpful for us to polish the paper. All typos were corrected in the version.

---

### Official Review · AnonReviewer4 · 2020-10-29
**The authors present a network agnostic framework for student teacher training paradigm. The experiments and results are presented for medical imaging datasets, where annotations are hard to achieve.**

**Rating:** 7
**Confidence:** 4

**Review:**

In this work the authors propose to transfer knowledge between teacher and student networks trained on separate datasets, and claim to overcome challenges in availability of data annotations for challenging semantic segmentation in medical imaging domain.

Strengths
The proposed model is simple to follow and is targeted towards a significant problem in medical imaging analysis domain.

Comments
The authors have used five segmentation networks, it is suggested that the selection of these five algorithms is further justified.
One of the major concern in medical imaging domain is the black-box nature of the DL algorithms used, authors should comment on how relying on this black-box nature for knowledge transfer would effect the interpretability of these results.
The method relies on three datasets and three models to come up with the final target segmentation, what are the requirements on the size of these datasets, in general the authors should discuss the effect of this on the overall performance.

---

> ### Author Response · Authors · 2020-11-23
> **Response to Reviewer 4**
>
> We really appreciate the encouraging comments. To better present the study, we have revised and polished the paper extensively.
>
> ~~~
> Reviewer's comment: The authors have used five segmentation networks, it is suggested that the selection of these five algorithms is further justified.
> ~~~
>
> Please find our explanation in **Section 3** -- “We posit that if the teacher model is ineffective, a more capable student model can easily achieve similar performance; if the student model, however, is ineffective, the cause is not easily identified. It can be attributed to the knowledge transfer algorithm or inherent limitations of the student model itself. We experimented with five state-of-the-art deep neural networks, which have different architectures and have been shown to be the best on various segmentation tasks.”
>
> Also, we would like to briefly introduce each method:
> - DeepLabv3+ is one of Google’s latest and best performing semantic segmentation models.
> - U-Net has been the most widely used network for medical image segmentation.
> - AttU-Net is a modification of U-Net, where attention gates help the model focus on the target.
> - SDU-Net  has demonstrated better performance by using only 40 percent of the parameters in U-Net.
> - Panoptic-FPN merges semantic segmentation and object detection, which provides a more rich and complete segmentation.
>
> ~~~
> Reviewer's comment: One of the major concern in medical imaging domain is the black-box nature of the DL algorithms used, authors should comment on how relying on this black-box nature for knowledge transfer would effect the interpretability of these results.
> ~~~
>
> Thank you for highlighting this important point.
>
> The main difference between our knowledge transfer algorithm and conventional deep learning training is that our transferal dataset is pseudo-annotated where the mask has no physical meaning. Interpretable machine learning techniques can be grouped into two categories: local interpretability and global interpretability. Local interpretability examines an individual prediction of a model locally, trying to figure out why the model makes the decision it makes. Global interpretability implies that the user can understand how the model works globally by inspecting the structures and parameters of a complex model.  Since local interpretability does not need the original training dataset, our knowledge transfer does not complicate the interpretability. However, it may complicate the global interpretability, as the parameters of the neural network are closely related to the training dataset. It would be interesting to study how interpretability is affected by knowledge transfer.
>
> Due to page limitation, we condensed the discussion in **Section 6** -- “Medical imaging applications requires interpretable algorithms which generate confidence in the clinical user. Knowledge transfer employs pseudo annotation for training which has no physical meaning. It is imperative to examine and quantify the interpretability of student model before deploying models clinically.”
>
> ~~~
> Reviewer's comment: The method relies on three datasets and three models to come up with the final target segmentation, what are the requirements on the size of these datasets, in general the authors should discuss the effect of this on the overall performance.
> ~~~
>
> Out the three datasets, only the pseudo-annotated transferal dataset is novel in this study. So, our discussion and revision mainly focuses on the transferal dataset. Please find our explanation below:
>
>  - **Teacher-training dataset:**  the teacher-training dataset is used to train the teacher model, which is the conventional training with ground truth. Our algorithm aims to solve the issue of knowledge transfer when the teacher-training dataset is not accessible; as such, the size of teacher-training dataset is not the primary focus of this study.
>
>  -  **Transferal dataset:**  As part of our revision, we detailed our method to refine the transferal dataset by excluding images without salient targets in the pseudo mask (in **Section 2**). Our experiments found that smaller datasets, wherein images without salient targets in the pseudo annotation are excluded, could result in better performance.  Table 2 and Table 3 demonstrate that all student models trained on pseudo-annotated Baheya Breast Lesion resulted in Dice scores similar to the teacher model, although the size of Baheya Breast Lesion is much smaller than the teacher-training dataset. We condensed the discussion in **Section 6** --“Transferal dataset with a large number of images of various contents has higher possibility to capture rich targets in the pseudo masks, but it may also include many images without salient targets. Future work includes optimization of a single transferal dataset or combine multiple transferal datasets to build a better one".
>
>  - **Fine-tuning dataset:** Similarly, the fine-tuning dataset is similar to that of conventional fine-tuning.

---

### Author Response · Authors · 2020-11-22
**Summary of Revision**

We sincerely thank the reviewers for their valuable feedback and insight. We have carefully  and extensively revised and polished the paper. We hope that our response will fulfill the desired alterations.

The major revisions are summarized below.

**Introduction:** We introduced knowledge distillation algorithms to contrast our method within the space of knowledge distillation and student-teacher training.

**Algorithm:** We reorganized the algorithm section. The main stay of our algorithm is knowledge transfer (Algorithm 1). However, if a downstream task is accompanied with a small dataset with ground truth, our knowledge transfer can be used together with fine-tuning (Algorithm 2 in Appendix). We make clear distinctions between the two algorithms and describe them in more details.

**Experiment:** We excluded the experiment on "direct learning" which lacked relevance for knowledge transfer. Meanwhile, we added 1) the experiment on knowledge transfer that only uses images with salient targets in the pseudo mask, and 2) the experiment on knowledge transfer from multiple weak teachers. We also defined and calculated the "knowledge transfer capability", which measures the capability of a transferal dataset to transfer the knowledge from a teacher model to a student model.

**Discussion:** We added the discussion section as suggested by the reviewers and provided our insights into the study.

**Key terms:** We changed some key terms for ease of understanding.

- "dataset agent" --> "transferal dataset"
- "educated student" --> "trained student"
- "pseudo annotation" --> "pseudo mask"

---

### Decision · Program_Chairs · 2021-01-07
**Final Decision**

**Decision:**

Reject

**Comment:**

A majority of the reviewers find the paper lacks novelty and provides an insufficient discussion of the state-of-the-art in knowledge distillation and student teacher training to warrant publication.
The approach is quite narrow to the application domain and the paper does not provide novel insights on how to chose a good network.
A subset of the experiments performed on an internal data set with random train-test-splits do not evaluate a realistic transfer setting.